# Relative Risk Perceptions between Snus and Cigarettes in a Snus-Prevalent Society—An Observational Study over a 16 Year Period

**DOI:** 10.3390/ijerph16050879

**Published:** 2019-03-11

**Authors:** Karl Erik Lund, Tord Finne Vedoy

**Affiliations:** Department Alcohol, Tobacco & Drugs, Norwegian Institute of Public Health, P.O. Box 222 Skøyen, N-0213 Oslo, Norway; tord.vedoy@fhi.no

**Keywords:** smokeless tobacco, snus, risk perception, tobacco, relative risk perception, e-cigarettes, NRT

## Abstract

*Background:* Most studies on perceived risks of smokeless tobacco products (SLT) have been conducted in the U.S., and the vast majority conclude that perceptions of the relative harmfulness of SLT versus cigarettes in the population are inconsistent with epidemiologically-based risk estimates, and typically conflated to that of cigarettes. We tested whether such inaccuracies also existed in Norway, where SLT-products are less toxic (Swedish snus) and SLT use is more prevalent than in the U.S. *Methods:* Over a 16 years period (2003–2018), 13,381 respondents (aged 16–79 years) answered questions about risk perceptions in Statistics Norway’s nationally representative survey of tobacco behavior. We applied an indirect measure of comparative harm where risk beliefs for eight nicotine products were assessed independently of other products and then compared the answers. The wording of the question was: “We will now mention a variety of nicotine products and ask you how harmful you think daily use of these are. Use a scale from 1 to 7, where 1 is slightly harmful and 7 is very harmful”. Mean scores with 95% confidence intervals were calculated. *Results:* The overall risk score for cigarettes was 6.48. The risk score for snus was 5.14–79.3% of the risk score of cigarettes. The relative risk scores for e-cigarettes (3.78) and NRT products (3.39) was 58.4% and 52.3% when compared to cigarettes. Perceptions of risk were stable over time. A strong association was observed between perceived risk of snus and having used snus in attempts to quit smoking. *Conclusion:* Perceptions of relative risk between snus and cigarettes is inconsistent with estimates from medical expert committees, which assess the overall health risk from use of Swedish snus to be minor when compared to the risk from smoking. Like the situation in the US, incorrect beliefs about SLT risks seem to be prevalent also in the snus-prevalent Norwegian setting. Accurate information on differential harms needs to be communicated. Future research should try to identify reasons why health authorities in the US and Scandinavia allow these well-documented misconceptions to persist.

## 1. Background

Tobacco and nicotine products fall on a continuum of risk based on the mode of nicotine delivery and product constituents. Consumers’ perceptions of the risk from using these products—subjective judgements about the potential harm to health—are important constructs in decision making theory [1,2,3] and are associated with numerous tobacco-related health behaviors, such as initiation [4,5], cessation [6,7,8] and product switching [9,10].

In the literature, consumers’ perceptions of risks are often reported as comparisons across the nicotine product spectrum with combustible cigarettes—the most hazardous form of nicotine uptake—as a reference category [11]. In addition to cigarettes, studies typically include risk assessments for nicotine replacement therapy products (NRT), e-cigarettes and various types of oral smokeless tobacco products (SLT). A recent systematic review of relative risk perceptions (RRP) across nicotine products identified 29 studies in which RRP of SLT products had been compared to combustible cigarettes [12]. The vast majority of these studies had been conducted in the U.S., and the authors concluded that perceived risks of SLT use were overestimated and conflated to that of cigarettes. Another U.S. study published after the review supported this finding [13].

In this study, we identify how Norwegians perceive the risk difference between SLT and cigarettes. This difference might be larger in Norway than what is typically reported from studies conducted in the U.S. Firstly, in comparison to other SLTs, the Scandinavian types of SLT—often referred to as Swedish snus—is acknowledged by the scientific community to be at the lower end of the risk scale in terms of adverse health effects [14,15,16]. The U.S. types of SLTs are manufactured, stored and used differently than is Swedish snus. During production, Swedish snus undergoes heat pasteurization to reduce formation of carcinogenic tobacco-specific nitrosamines (TSNAs) and polycyclic hydrocarbons (PAHs) [17].

Secondly, risk beliefs might be better anchored in Norway than in the U.S. In the U.S., the awareness and prevalence of SLT use is low [18,19] compared to the situation in Norway where there currently are as many snus users as smokers in the population, and the snus phenomenon has been the subject of massive media attention [20]. Thus, the epistemological climate for making judgements about relative harmfulness should be favorable and reduce confusion. Knowledge and use of a nicotine product is often associated with more accurate risk beliefs [12,21].

Thirdly, the majority of studies included in the review [12] employed a direct measure of comparative harm, which elicits estimates of relative risks of SLT and cigarettes within a single question (e.g., compared to regular cigarettes, are SLTs less harmful, as harmful or more harmful). Evidence suggests that using an indirect method, as we do—measuring the perceived harm of SLT and cigarettes in two separate questions and then comparing the answers—will increase the proportion of respondents who define SLTs to be less harmful than cigarettes [10,22,23,24,25].

### Aims

In this study, we compare risk perceptions of snus use and smoking as reported by respondents living in a snus-prevalent environment. More precisely, we have examined whether relative risk perceptions of snus and cigarettes in the general population—and five other nicotine products—have changed over a 16-year period characterized by a decline in smoking and an increase in snus use (aim 1). Secondly, we singled out the risk perceptions of snus that were most inconsistent with risk estimates from expert committees, and then studied how this overestimation of risk was associated with respondent characteristics (aim 2). Third, we examined the association between perceived harmfulness of snus and the likelihood of using the product in quit-smoking attempts (aim 3).

## 2. Methods

### 2.1. Material

Data stem from Statistics Norway’s nationally representative survey of tobacco behavior among adults (16–79 years) conducted annually since 1973. Questions about risk perceptions were included each year in the period 2003–2011, excluded from 2012 to 2016, but reintroduced in 2017. In the period 2003 to 2014, a sample of 2000 Norwegians were contacted every fourth quarter by telephone and asked questions about their past, present and planned tobacco use, attitudes towards tobacco preventive measures, exposure to passive smoking, etc. In the period 2015 to 2018, the sample size was increased to 3000. The average response rate was 62% for surveys conducted from 2003 to 2011 and 58% for 2017–2018. In total, 13,381 respondents were included in our study.Surveys conducted by Statistics Norway are carried out in line with the European Statistics Code Of Practice. Statistics Norway states to respondents that participation in the survey is voluntary and that respondents can withdraw from the interview at any time. The survey is anonymous and the participants cannot be traced. The data that support the findings of this study are available from Statistic Norway, but restrictions apply to the availability of these data, as they were used under license for the current study. However, the data are available from the authors upon reasonable request and with permission of Statistics Norway.

### 2.2. Measures

We used an indirect measure of perceived relative harm. The wording of the question was: “We will now mention a variety of nicotine products and ask you how harmful you think daily use of these are. Use a scale from 1 to 7, where 1 is slightly harmful and 7 is very harmful”. Combustion products included conventional manufactured cigarettes, manufactured cigarettes with reduced nicotine and tar, roll-your-own cigarettes, cigars/cigarillos and pipe. The latter three were not included in the 2017/18 surveys. Non-combustible products included Swedish snus, NRT and e-cigarettes (ENDS). The latter was only included in the 2017 and 2018. Respondents who ticked off a “don’t know”-category, were excluded from the analysis.

### 2.3. Current Use of Products

Respondent’s smoking status was classified in five categories: never-smoker, former smoker (daily or occasionally), current occasional smoker, current daily smoker (non-hard-core) and “hard-core” smoker. The latter included daily smokers who had a) made no quit attempts, b) had no plans to quit smoking, and c) who portrayed themselves as smokers in five years’ time [26]. Snus use was defined using similar categories, but did not include a “hard-core” category.

### 2.4. Socio-Demographic Information

Respondents’ socio-economic position was measured using education as a proxy and divided into three groups. Respondents who had completed nine years of compulsory education (primary), respondents who had completed at least three years of high-school education (secondary) and respondents who had completed at least three years of university education (tertiary). Age was categorized into four groups (16–24, 25–44, 45–66 and 67–79 years).

## 3. Analysis

### 3.1. Aim 1: Risk Perceptions of Nicotine Products and Changes Over Time

For each year as well as for all years combined, we calculated mean risk score on the 7-point Likert scale and 95% confidence intervals. To examine how respondents perceived each nicotine product relative to conventional cigarettes, we calculated relative harm scores by dividing the overall score (all years combined) for each product with the score for conventional cigarettes ×100.

### 3.2. Aim 2: Association of Snus Risk Overestimation with Respondent Characteristics

We constructed three nested logistic regression models with a dichotomous measure of snus risk perception in which the values 6 and 7 denoted high perceived risk and all other values (1–5) denoted medium or low perceived risk. In model 1, perceived risk of snus (high vs medium/low) was regressed on smoking status alone. Model 2 also included information on education, sex, age and survey year, all coded as dummy variables. Lastly, model 3 included additional information on snus use. To examine what background characteristics were important for perceiving snus use as very harmful, we calculated average adjusted probabilities (and 95% confidence intervals) of rating the risk from using snus as 6 or 7, from models 1–3, using the margins command in Stata 15. Both AIC (Akaike information criterion) and BIC (Bayesian information criterion) strongly suggest that model 3 fit the data best.

### 3.3. Aim 3: Relation between Perceived Harmfulness of Snus and the Likelihood of Using Snus in Quit-Smoking Attempts

Lastly, we constructed a fourth logistic regression model in which the dependent variable defined ever smokers (current or former) who had used snus in their latest quit attempt (*N* = 691/4924). Independent variables were perceived risk of snus use, education, sex, age and survey year. In addition, the model included an interaction between smoking status and all other independent variables to check if the association between perceived risk of snus and using snus in the last quit attempt differed between former smokers (successful quitters) and current smokers (unsuccessful quitters). Results are presented as adjusted predicted probabilities.

## 4. Results

Our sample had a mean age of 45.3 years. Other demographic, smoking and snus use characteristics are described in Table 1.

Figure 1 illustrates the inverse correlation between smoking and snus use in the adult Norwegian population for the period 2003–2018.

Figure 2 illustrates that the perceived risks have been stable for all nicotine products over this 16-year period. Combustible nicotine products are generally perceived as more hazardous than non-combustible products. When the risk score for cigarettes (6.48, SD 0.88, all survey years combined) is set to 100%, the risk score for snus (5.14, SD 1.48, all years combined) makes up 79.3% relative to cigarettes (not shown in the figure). In other words, daily snus use is perceived to be 79.3% as harmful as daily cigarette smoking. The corresponding relative risk scores for E-cigarettes (3.78, SD 1.67, 2017 and 2018 combined) and NRT products (3.39, SD 1.58, all years combined) were 58.4% and 52.3% respectively.

If we examine current smokers and current snus users separately, we find that perceptions are stable over time, but that both groups generally perceive use of tobacco products as less harmful than the general population (not shown). In contrast, compared to the general population, snus users seem to regard their own snus use as much less harmful (3.93 vs 5.14), whereas smokers rate smoking as marginally less harmful (6.29 vs 6.48).

The overall likelihood of perceiving snus as very harmful (score 6 or 7) was 0.40 (Table 2). Based on model 3 we find that being female (0.49) and being in the oldest age group (0.46) increased the probability to classify snus as very harmful.

Smokers, in particular hard-core smokers (0.33), had lower probability of perceiving snus as very harmful compared to never-smokers (0.42) (model 3). The lowest probability of perceiving snus as very harmful was observed among daily snus users (0.14), occasional snus users (0.20) and former snus users (0.29) (model 3). Moreover, inclusion of snus use in Model 3 did not affect the association between smoking status or sex, and snus risk perception, which indicated that the differences between men and women, and between smoking status groups, were not a result of the known differences in snus use patterns.

Figure 3 shows the results from the fourth logistic regression model and illustrate the strong association between perceived risk of snus and having used snus at the lasted attempt to quit smoking. This association was almost identical among former smokers (successful quitters) and current smokers (unsuccessful quitters). Testing the differences between adjacent coefficients showed that the probability of having used snus in the latest quit attempt was lower for each additional level of snus risk perception, both among current and former smokers (*p* < 0.05), with the exception of the two lowest values of risk perception (1 and 2).

## 5. Discussion

Our study has shown that in the Norwegian population, where use of Swedish snus is widespread, there is a persistent belief that the risk of daily snus use is equivalent to nearly 80% of the risk from daily smoking. The assumed risk difference between the two products is very inconsistent with estimates from medical expert committees, which typically assess the overall health risk from use of Swedish snus to be minor (below 10%) when compared to the risk from smoking [15,27,28,29,30]. Incorrect beliefs about SLT (smokeless tobacco) risks seem to be prevalent also in the snus-prevalent Norwegian setting as also observed in the U.S. where SLT products are more toxic and use is uncommon.

Moreover, our findings reveal that the overestimations of the harm of snus have been consistent over time. This apparent stability obscures two underlying but counteracting mechanisms: A growing inflow of snus users (Figure 1) who generally rate snus lower on the risk scale (Table 2), pulled the overall risk estimate for snus downwards. At the same time, the relatively large group of never users of snus perceived snus to be moderately (but significantly) more hazardous over time (result not shown), thus pulling the overall estimate upwards.

Despite inaccurate RRPs, there has been a dramatic transition from cigarettes to snus in Norway since 2003 (Figure 1). This could perhaps have been even stronger if RRP had been more in line with medical consensus, even if some snus users correctly believed snus is less harmful when they were still smokers, and this may have helped to motivate their switch. Other contributing factors for the marked shift might be the price difference between the two products (which is quite narrow), the implementation of increasingly stricter indoor smoking regulations in a country with a harsh climate for outdoor smoking, and the growing anti-smoking sentiment that follows policy implementation. The symbolic meaning of snus use is not considered in the same negative way as smoking in Norway [31,32], and may have promoted snus use, even if risk perceptions are stable. Admittedly, however, little is known about such possible causes. The most important factor is that use of snus has become the most frequent method to quit smoking in Norway [33], (after unassisted quitting). The important role of snus in smoking cessation has been addressed in several studies in Norway [34,35,36,37,38].

### 5.1. Snus vs E-Cigarettes

In contrast to e-cigarettes, a heterogeneous product with short history, Swedish snus is a standardized product with a long history. For Swedish snus, a convincing epidemiological evidence base shows no associations with lung cancer or respiratory diseases, and very weak, if any, association with cardiovascular diseases [15,16]. These diseases combined cause 2/3 of the smoking attributable mortality. Moreover, there is no evidence that use of Swedish snus is associated with any major health hazard that does not also arise from tobacco smoking [15].

In contrast, long-term epidemiological studies for e-cigarettes are not yet available. Claims of risk reduction are currently based on toxicological testing, animal studies and acute physiological reactions in humans. The heterogeneity of the products makes risk assessments even more complicated. Despite a more robust evidence base for providing a lower estimate of risk, Swedish snus (score 5.14) was perceived as more harmful than e-cigarettes (score 3.78).

The explanation might be that the risk concept for snus also include a concern for the much-discussed increase in snus use among youth, which the health authorities have labeled an epidemic. In the age-group 16–24 years, the prevalence of daily snus use was 24.7% among men and 15.9% among women in 2017, an increase from 11.9% and 1.1% in 2003 respectively [39]. In contrast, the prevalence of vaping has been below 2% [40], and e-cigarettes have consequently not been portrayed as a threat to public health. Moreover, in 2016, the Government decided to lift the ban on the sale of nicotine containing e-liquids, and this might have downgraded the perceptions of risk from e-cigarettes further.

In the U.S. the situation is inverse. The increase in vaping has been labeled an epidemic by the health authorities, and the prevalence of SLT use is low. Nevertheless, also in the U.S. SLT products seem to be perceived with greater harm than e-cigarettes [13].

### 5.2. Risk-Use Equilibrium

It may appear tolerable to accept that never-smokers overestimate the harm from snus use, as this might protect some from taking up a product not totally risk-free. Misconceptions among smokers, in particular hard-core smokers, is perhaps harder to accept because inaccurate beliefs might be a barrier for a switch to less hazardous uptake of nicotine. In our study, we note that never-smokers tend to overestimate the risk from snus more than smokers (Table 2). The question is whether dissemination of information to correct RRP will encourage snus use among never-smokers so much that the reduced risk for smokers who swap cigarettes for snus (as a consequence of being better informed) sums to greater risk for the entire population.

To evaluate the possible problems caused by trying to correct beliefs, it is helpful to consider what might be called the risk/use equilibrium—an equilibrium achieved by increasing use as risk decreases [41]. Given the relatively low excess risk for never smokers who take up snus, and given the epidemiological verified huge risk reduction for smokers who switch to snus, the number of never-smokers taking up snus must go to unlikely large proportions to balance out the health gain from the smokers who switch. For net harm to occur, an epidemiological modelling study estimated that 14–25 ex-smokers would have to start using snus to offset the health gain from every smoker who switched to snus rather than continuing to smoke. Likewise, 14–25 people who have never smoked would need to start using snus to offset the health gain from every new tobacco user who used snus rather than smoking [42].

Up until now, studies have demonstrated that the largest reservoir of potential snus users has been smokers, and that use among never-smokers is small [43]. Based on current knowledge of the a) absolute risk of snus use relative to non-use, b) the relative risk of snus relative to smoking, and c) how the population of snus users is made up of never-smokers and smokers (the user configuration), dissemination of information to correct misconceptions of relative risk will probably result in a net gain to public health.

### 5.3. Confirmation and Optimism Bias

Consistent with other studies, being a current user of snus was associated with a greater likelihood of endorsing the view that snus was less harmful than cigarettes relative to non-users of snus [44,45,46]. These findings may be explained by theories of selective exposure and perception [47] or optimism bias [48]. An alternative interpretation could be that snus users in Norway, the vast majority being ever-smokers [43], themselves have experienced significant improvements in health after switching products. In any case, the cross-sectional nature of the study precludes any examination of the temporal relationship between user perceptions and behavior.

### 5.4. Limitations

There is currently no consensus on how to best measure tobacco risk perceptions in tobacco control research. Two reviews of risk perception measurement recommended assessing perceptions of specific tobacco-related outcomes [11,49] rather than general perceptions of harm, as we did in our study. However, one study from Norway [50] assessed perceptions of the relative snus/cigarette risk of four specific outcomes; cardiovascular disease, cancer of the lung, stomach, and oral cavity. For all diseases except lung cancer, the majority wrongly thought snus users were running a higher or equal risk. For lung cancer, 22% believed that snus use gave a higher or equal risk. The authors concluded, in line with our finding, that erroneous ideas of approximately equal harm from snus and cigarettes were common in the general population.

In our study we make use of parametric statistics (mean score) and express perceived relative harm as a percentage of the mean scores for each product compared to the score for cigarettes set at 100%. This implies that we are treating ordinal variables, originating from a 7-point Likert scale, as if they were continuous. This is quite common in the social sciences in general and in attitude research in particular [51]. However, each value on our Likert scale is anchored to an interpretation, and the numerical distance between each set of subsequent interpretations is not necessarily equal. In spite of being widely used, applying parametric statistics on ordinal variables continue to be a matter of dispute among methodological purists.

### 5.5. Strengths

The study used national representative samples applying identical measures of risk over a long time period. Risk perceptions were examined across multiple products therefore allowing comparisons of the relative perceptions of the harmfulness of different products. Response options had verbal qualitative comparisons ranging from slightly harmful to very harmful, avoiding numerical estimates of risk that have proved difficult to use in lay people [52]. We used indirect measures of risk that may help reduce social desirability bias, particularly in contexts where the ‘no safe tobacco product’ message resounds [12] as in Norway. However, other research suggests indirect measures may have lower validity and or provide any new information beyond what can be extracted from using direct measures [25]. While most other studies on risk perceptions of nicotine products are descriptive [12], we also examined how perceptions are associated with product use and use of snus in smoking cessation.

Only 1% and 2% of the respondents reported that they did not know the health risk from smoking and snus use respectively, supporting our assumption that most Norwegians are familiar with the products. In the analyses, these respondents were excluded. To check the effect of omitting these respondents, we reran all analyses with these respondents coded as 3.5. This did not alter the results. In addition, to examine a possible urban-rural dimension in snus risk perception, we included a measure of centrality in model 3. This variable was not associated with differences in risk perception and did not alter any of the other estimates, and was therefore dropped from the model.

### 5.6. Implications

The World Health Organization (WHO) has emphasized that the public has a right to accurate information on the harms of tobacco and that “countries have a legal obligation to provide it” [53]. However, independent of risk perception measurement method (indirect vs direct), outcome (general harm vs specific diseases), and region (U.S. vs Scandinavia), the vast majority of studies (including ours) demonstrate that the health risk posed by smokeless products is consistently overestimated relative to the risk of smoking cigarettes.

The lay public often simply parse things into ‘safe’ or ‘unsafe’ [54]. This has been referred to as the law of contagion [55]. Research indicates that individuals tend to convey the message “Snus is not a safe alternative to cigarettes” that snus is not a safer alternative to cigarettes (i.e., just as harmful as cigarettes) [22,56,57]. The health warning printed on snus boxes in Norway has been “Snus can damage your health and is addictive”, but according to EU-regulations the modal verb can will eventually be removed. A recent study from Norway demonstrated that risk information material from the health authorities strongly distorted risk perceptions for snus in relation to cigarettes and raised already exaggerated risk beliefs [58]. This finding was in line with one of the conclusions from the systematic review: *“Individuals may have been misled by public health authorities, which have provided misinformation on the risk of SLT relative to combustible cigarettes in the USA, where most of the reviewed population surveys were conducted”* [12].

### 5.7. Future Research

We argue that a timely and relevant question for future qualitative research should be to identify reasons why health authorities in the U.S. and Scandinavia allow these well-documented misconceptions to persist. Perhaps they feel they lack encouragement from researchers? Even after uncovering severe misconceptions of relative risk in their respective studies, several researchers refrain from encouraging health authorities to correct misbeliefs about relative risks e.g., [13,24,59]. Perhaps moral psychological issues are influencing the reluctance from health authorities to inform on differential risks, as some have suggested [60].

Health authorities might perhaps also have concerns about potential negative consequences if consumers’ misbeliefs were to be corrected. If this is the case, such logic will prompt a range of associated follow-up questions: Should fears of the possibility of negative consequences for some persons justify suppressing or omitting health relevant information, especially when there is no actual evidence that there would be any net negative effects on population health? Should public health considerations prevail the principle of informed decision? Should the effectiveness of risk communication be measured by the extent to which it is able to move recipients to specific behavioral outcomes that the sender find desirable (e.g., continued nicotine abstinence in never-smokers)? Alternatively, maybe risk communication instead should be judged by how it is interpreted by the recipients?

Some researchers have argued [61] that quarantining information on differential risks need to be justified, not merely by fears of net negative public health effects, but by convincing evidence that such measures are actually warranted, that public health overall is in imminent danger and that the danger is sufficient to override principles of individual autonomy. In countries where tobacco/nicotine products are legally sold, and also differ greatly in disease risks compared to cigarettes (e.g., snus and e-cigarettes), science-based, comprehensible, and actionable health information on differential risks should be available and only reconsidered if it is established that this information is causing losses to population health overall.

## 6. Conclusions

While continuing to emphasize the extreme lethality of smoking, accurate information on differential harms needs to be communicated when snus is legally available. The extent of inaccurate health beliefs regarding these products, as demonstrated in this and most other studies on RRP, implies that significant efforts will be required to bring the public’s knowledge in line with epidemiologically based evidence. Irrespective of the anticipated difficulty involved in constructing relative health-risk messages and irrespective of the fear of misinterpretations by some persons, the health authorities should work to facilitate better informed choices by tobacco consumers. Such public health information should not be left in the hands of an industry marketing products to simply promote sales.

## Figures and Tables

**Figure 1 ijerph-16-00879-f001:**
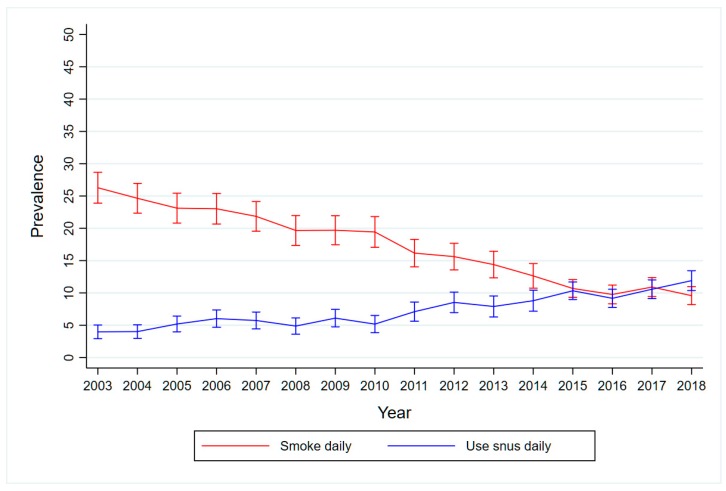
Unadjusted prevalence of daily smoking and daily use of snus among Norwegian adults ages 16–79 years, 2003–2018.

**Figure 2 ijerph-16-00879-f002:**
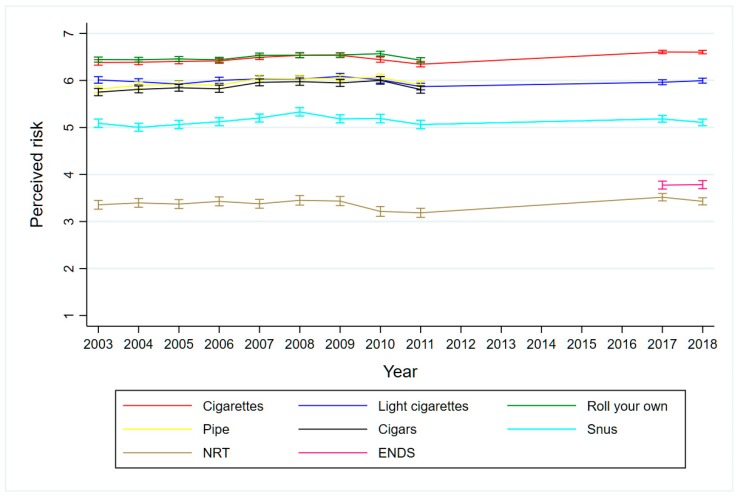
Perception of the harmfulness of eight nicotine products among Norwegian adults ages 16–79 years, 2003–2018. Unadjusted mean risk scores on a scale from 1 (slightly harmful) to 7 (very harmful).

**Figure 3 ijerph-16-00879-f003:**
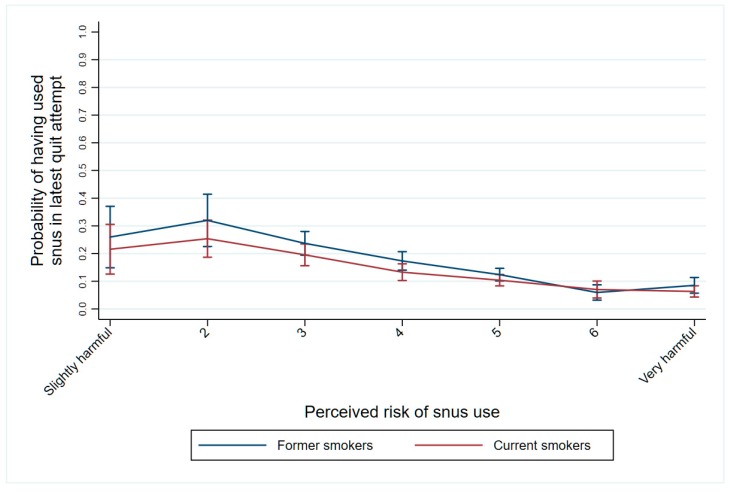
Adjusted predicted probabilities of having used snus in the latest attempt to quit smoking at different levels of perceived risk of, among current smokers (*n* = 381) and former smokers (*n* = 310).

**Table 1 ijerph-16-00879-t001:** Sample description, *N* = 13,381.

Variable	Percent	Standard Deviation
Smoking status		
“Hard core” smoker	5.6	23.1
Daily smoker, not “hard core”	13.4	34.0
Occasional smoker	7.6	26.5
Former smoker	29.5	45.6
Never smoked	44.0	49.6
Snus use status		
Use snus daily	6.9	25.4
Use snus occasionally	4.0	19.7
Used snus previously	5.8	23.4
Never used snus	83.3	37.3
Education		
Primary	19.3	39.5
Secondary	47.8	50.0
Tertiary	32.8	47.0
Sex		
Men	50.1	50.0
Women	49.9	50.0
Age		
16–24	12.8	33.4
25–44	36.6	48.2
45–66	38.2	48.6
67–79	12.4	33.0
Snus risk perception		
Rated snus 1 to 5 of 7	60.2	48.9
Rated snus 6 to 7 of 7	39.8	48.9
Cigarette risk perception		
Rated cigarettes 1 to 5 of 7	13.0	33.6
Rated cigarettes 6 to 7 of 7	87.0	33.6

**Table 2 ijerph-16-00879-t002:** Adjusted predicted probabilities (Pr) of perceiving snus as very harmful (score 6 or 7 on a scale from 1- slightly harmful to 7-very harmful) across tobacco use status, education, sex, age and survey year.

	Model 1	Model 2	Model 3
	Pr (95% CI)	Pr (95% CI)	Pr (95% CI)
Overall probability	0.40 (0.39–0.41)	0.40 (0.39–0.41)	0.40 (0.39–0.41)
Smoking status			
“Hard core” smoker	0.34 (0.31–0.37)	0.33 (0.30–0.36)	0.33 (0.30–0.36)
Daily smoker, not “hard core”	0.40 (0.37–0.42)	0.39 (0.37–0.41)	0.39 (0.37–0.41)
Occasional smoker	0.33 (0.30–0.36)	0.37 (0.34–0.40)	0.40 (0.37–0.43)
Former smoker	0.39 (0.37–0.40)	0.38 (0.36–0.39)	0.39 (0.37–0.40)
Never smoked	0.42 (0.41–0.43)	0.43 (0.42–0.44)	0.42 (0.40–0.43)
Education			
Primary		0.42 (0.40–0.44)	0.42 (0.40–0.44)
Secondary		0.41 (0.40–0.42)	0.41 (0.40–0.42)
Tertiary		0.37 (0.35–0.38)	0.36 (0.35–0.38)
Sex			
Men		0.29 (0.27–0.30)	0.30 (0.29–0.31)
Women		0.51 (0.50–0.52)	0.49 (0.47–0.50)
Age			
15–25		0.26 (0.24–0.28)	0.29 (0.26–0.31)
25–44		0.38 (0.37–0.40)	0.40 (0.39–0.41)
45–66		0.43 (0.41–0.44)	0.41 (0.40–0.42)
67–80		0.50 (0.47–0.52)	0.46 (0.44–0.48)
Survey year			
2003		0.41 (0.39–0.44)	0.40 (0.38–0.43)
2004		0.36 (0.34–0.39)	0.36 (0.33–0.38)
2005		0.39 (0.36–0.41)	0.38 (0.35–0.41)
2006		0.38 (0.36–0.41)	0.38 (0.35–0.40)
2007		0.40 (0.38–0.43)	0.40 (0.37–0.43)
2008		0.45 (0.42–0.48)	0.44 (0.41–0.47)
2009		0.42 (0.39–0.45)	0.42 (0.39–0.45)
2010		0.39 (0.36–0.42)	0.39 (0.36–0.42)
2011		0.37 (0.34–0.40)	0.37 (0.35–0.40)
2017		0.41 (0.39–0.43)	0.43 (0.40–0.45)
2018		0.39 (0.36–0.41)	0.40 (0.38–0.43)
Snus use			
Use snus daily			0.14 (0.11–0.16)
Use snus occasionally			0.20 (0.16–0.24)
Used snus previously			0.29 (0.26–0.33)
Never used snus			0.43 (0.42–0.44)
BIC	17,991	17,137	16,808
AIC	17,953	16,980	16,628
*N*	13,381	13,381	13,381

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
