# Peer review of "Relative Risk Perceptions between Snus and Cigarettes in a Snus-Prevalent Society—An Observational Study over a 16 Year Period"

_ijerph, 2019, doi:10.3390/ijerph16050879_

Reviewer 1 Report

This study uniquely examines snus and cigarette risk perceptions in large national surveys over a long period of time, in a country which has experienced increases in snus use over this same time period - Norway. As such this paper has the potential to make a nice addition to the literature on this topic. However I have several questions and comments.

On line 52 of the introduction it says that authors hypothesize that the perceived risks are more pronounced in Norway than North America – first it is not clear what is meant by “more pronounced” but also, this sentence seems to suggest that this study will compare data across these regions when it does not actually do so.

Table 1 - Some of the age categories are different than what is mentioned in the methods (should table 1 say 16-24 rather than 15-25? And 67-79 rather than 67-80?

Also seems likes the overall risk perception for cigarettes could also be added to Table 1 as is for snus. What % rated cigarettes as a 6 or 7? How does that compare to the level for snus (40%)?

I’m not sure about the author’s calculation of relative harm scores – it is hard to interpret. What does a score of 79% mean? Saying that the “risk score snus makes up 79.3% relative to cigarettes” is not a clear statement. Are you saying that daily snus use is perceived to be 79% as harmful as daily cigarette smoking?  I’m not sure about the accuracy of this way of presenting a perceived risk comparison.

An alternative  way might be to compare the odds with which participants rated cigarettes as being very harmful (ie, 6 or 7) compared to doing so for snus and for the other products?

More results narrative could be used to describe the results in Table 2.

Figure 2 is interesting but only shows risk perceptions among the entire population. Did authors also look to see if risk perceptions changed or were stable among smokers specifically? To see if this might correlate with the change in use trends observed in Figure 1? Ultimately it seems it is the perceptions of snus among current smokers that is particularly important from the perspective of motivating switching and harm reduction. The second paragraph of the discussion seems to suggest that authors may have looked at the results by some subgroups.

Figure 3 seems to show a trend between perceived risk of snus use and probability of having used snus in the latest quit attempt, but its not clear if this trend is statistically significant.

Discussion: The first paragraph of the discussion states that incorrect SLT risk perceptions seem to be equally prevalent in Norway as in the US and that no empirical support was found for authors’ initial hypothesis. This seems to be overstated given that authors did not actually compare risk perceptions between the two areas. It may be that the risks of SLT and snus are overestimated in both regions, but still less so in Norway than in North America. This is not clear from the paper….

Also the data from this paper show that over time, the overall mean perceived absolute harm from snus use has been perceived to be lower than that of cigarettes – this is correct (although perhaps the perceived absolute harm is higher than it should be). But authors don’t present the percentage of the population over time that have had more overt inaccurate perceptions – i.e., those who rate snus and cigarette smoking as equally harmful or snus as even more harmful. This might also be helpful to see and could be calculated using the indirect measures as has been done in some of the previous studies authors have cited (e.g., Popova & Ling)…

Under section 5.3 it could also be mentioned that some snus users correctly believed snus is less harmful when they were still smokers, and this may have helped to motivate their switch.

Finally, authors make a strong argument that there is a need to correct snus misperceptions and that researchers/authorities should actively inform the public about snus/cigarette relative risks. Yet, the data in this study seems to show that despite the consistent level of misperceptions in Norway over time, patterns of snus use went up. For some readers this may seem to undermine the importance of risk perceptions and of addressing misperceptions. Perhaps authors can comment on this.  

Author Response

Dear Editors,

Thank you for providing us with the opportunity to revise our article ijerph-446751 to the IJERPH. We are very grateful for two very constructive reviews. We have tried to revise the manuscript according to the reviewers’ comments. In the attached document we explain point-by-point the details of our revisions and offer some in-depth considerations to the reviewers' comments. In the manuscript we have highlighted all revisions with the "Track Changes" function in Microsoft Word. We hope the manuscript now will be processed further.

Reviewer 2 Report

This is a very interesting paper employing a major dataset.

The authors argue for the strengths of Indirect measures, but it should be clearer that arguments exist for both direct and indirect measures. These measures behave differently and encouragement has been offered to employ both types. For example, Persoskie et al (2017 (ref ) found that direct measures were more strongly associated with criterion variable: “For two of the three criterion variables (ever-trying e-cigarettes; SLT use status), models with both measures entered simultaneously revealed significant associations between the criterion variable and the direct measure but not the indirect measure. These results suggest that the indirect measures had some degree of validity but were outperformed by–and did not provide information beyond–the direct measures.“

It maybe that the magnitude of the risk difference is not as important as a there being a perceived difference (especially in a product they might enjoy and is widely available). As noted the proper ‘metric’ to employ is hard to know. Their use of percent differences in the indirect measures may impose a kind of structure on the findings that may not be compatible with the psychology of how risks are compared. Innumeracy is more likely to be a problem than not.  The consistent, clear-cut differences in risks as measured may be enough to help promote the rise in snus use.

Consider this earlier work: “Like other consumers, smokers do not possess an actuarial, numerically sophisticated appreciation of the risks of what they do, and for risk reduction, the thought that a product is “less risky” may be sufficient to influence behavior. Do those taking an aspirin each day do so because they have a numerical appreciation of their reduced risks or because they think they will simply be better off?” p1319 AJPH, August 2000, Vol. 90, No. 8, Kozlowski, Goldberg and Yost, ‘Measuring Smokers’ Perceptions of the Health Risks From Smoking Light Cigarettes’

I think it is likely the case that most members of the public do not spend time thinking about their risks of harm (is harm ‘death’ or ‘disability’)? Are snus users disproportionately worried about oral cancer? Do they have no fear of lung cancer?  Do they think about stroke?).  The harms from tobacco products are hard for experts to keep straight in a precise way.  The public at best has a crude sense of this, and we apply measurements that find reliable relationships but it is unclear ‘what’ they are actually measuring or on what scale it is really being measured.

The authors might consider also exploring how a difference score in risk predicts use of snus in quitting.

line 251, page 9 “However, one study from Norway [46] assessed perceptions of the relative snus/cigarette risk 251 of four specific outcomes; cardiovascular disease, cancer of the lung, stomach, and oral cavity.”  I cannot find this information in Ref 46 which refers to a study of GPs.

Author Response

Dear Editors,

Thank you for providing us with the opportunity to revise our article ijerph-446751 to the IJERPH. We are very grateful for two very constructive reviews. We have tried to revise the manuscript according to the reviewers’ comments. In the attached document we explain point-by-point the details of our revisions and offer some in-depth considerations to the reviewers' comments. In the manuscript we have highlighted all revisions with the "Track Changes" function in Microsoft Word. We hope the manuscript now will be processed further.

We also take the opportunity to reconfirm the existing author list and the corresponding affiliation.

Round  2

Reviewer 1 Report

The authors have been very response to reviewer’s comments and I appreciate their additional analyses. Thank you. In particular, authors’ additional analyses examining the estimated relative risk perceptions based on using the two indirect measures as presented in Table R4 is interesting and important. I appreciate that authors may want to save these additional analyses for another paper, and I hope they do so. However, given these results and the data that authors will leave in the paper (which are based on absolute harm measures), I feel that some of the language and statements in some parts of the discussion should be revisited, particularly with regard to references to “misperceptions” or “innacurate beliefs/relative risk perceptions”.

For example, I would recommend revising the first sentence of the second paragraph of the discussion section which says that “our findings reveal that misperceptions about relative risk in the population has been stable”. I don’t think the data presented in the study and the additional analyses authors presented in their response quite support this statement.  To me the data in the paper show that, over time, the estimates of perceived absolute harm from snus have been lower than the perceived absolute harm from cigarettes – I wouldn’t call this a misperception here because the harm of snus is lower than that of smoking. However, another way to get at authors point could be to state that the public generally seem to overestimate the harm of snus, and that these overestimations of the harm of snus have been consistent over time.

Author Response

Rewiever #1:

1) The authors have been very response to reviewer’s comments and I appreciate their additional analyses. Thank you.

Authors’ response (AR): Thank you for a very constructive first review.

2) In particular, authors’ additional analyses examining the estimated relative risk perceptions based on using the two indirect measures as presented in Table R4 is interesting and important. I appreciate that authors may want to save these additional analyses for another paper, and I hope they do so.

AR: We have already started additional analysis beyond those conducted in response to your first review and begun drafting a new manuscript based on these results.

3) However, given these results and the data that authors will leave in the paper (which are based on absolute harm measures), I feel that some of the language and statements in some parts of the discussion should be revisited, particularly with regard to references to “misperceptions” or “innacurate beliefs/relative risk perceptions”.

AR: Yes, we agree that would be a logical consequence.

4) For example, I would recommend revising the first sentence of the second paragraph of the discussion section which says that “our findings reveal that misperceptions about relative risk in the population has been stable”. I don’t think the data presented in the study and the additional analyses authors presented in their response quite support this statement.  To me the data in the paper show that, over time, the estimates of perceived absolute harm from snus have been lower than the perceived absolute harm from cigarettes – I wouldn’t call this a misperception here because the harm of snus is lower than that of smoking. However, another way to get at authors point could be to state that the public generally seem to overestimate the harm of snus, and that these overestimations of the harm of snus have been consistent over time.

AR: Yes, the suggested semantic change would absolutely be a more accurate representation of what data actually shows.

Old: Moreover, our findings reveal that misperceptions about relative risk in the population has been stable. This stability obscures two underlying but counteracting mechanisms: A growing inflow of snus users (Figure 1) who generally rate snus lower on the risk scale (Table 2), pulled the overall risk estimate for snus downwards. At the same time, the relatively large group of never users of snus perceived snus to be moderately (but significantly) more hazardous over time (result not shown), thus pulling the overall estimate upwards.

New section: Moreover, our findings reveal that the overestimations of the harm of snus have been consistent over time. This apparent stability obscures two underlying but counteracting mechanisms: A growing inflow of snus users (Figure 1) who generally rate snus lower on the risk scale (Table 2), pulled the overall risk estimate for snus downwards. At the same time, the relatively large group of never users of snus perceived snus to be moderately (but significantly) more hazardous over time (result not shown), thus pulling the overall estimate upwards.